# Environmental DNA reveals seasonal shifts and potential interactions in a marine community

Anni Djurhuus [1]*, Collin J. Closek[2,3]*, Ryan P. Kelly[4], Kathleen J. Pitz [5], Reiko P. Michisaki[5], Hilary A. Starks [2,3], Kristine R. Walz[5], Elizabeth A. Andruszkiewicz [3], Emily Olesin[6], Katherine Hubbard[6], Enrique Montes [1], Daniel Otis [1], Frank E. Muller-Karger [1], Francisco P. Chavez [5], Alexandria B. Boehm [3] & Mya Breitbart [1]*

Environmental DNA (eDNA) analysis allows the simultaneous examination of organisms across multiple trophic levels and domains of life, providing critical information about the complex biotic interactions related to ecosystem change. Here we used multilocus amplicon sequencing of eDNA to survey biodiversity from an eighteen-month (2015–2016) time-series of seawater samples from Monterey Bay, California. The resulting dataset encompasses 663 taxonomic groups (at Family or higher taxonomic rank) ranging from microorganisms to mammals. We inferred changes in the composition of communities, revealing putative interactions among taxa and identifying correlations between these communities and environmental properties over time. Community network analysis provided evidence of expected predator-prey relationships, trophic linkages, and seasonal shifts across all domains of life. We conclude that eDNA-based analyses can provide detailed information about marine ecosystem dynamics and identify sensitive biological indicators that can suggest ecosystem changes and inform conservation strategies.

[1] University of South Florida, College of Marine Science, 140 7th Avenue South, St. Petersburg, FL 33701, USA. [2] Stanford Center for Ocean Solutions, Stanford University, 473 Via Ortega, Stanford, CA 94305, USA. [3] Department of Civil and Environmental Engineering, Stanford University, 473 Via Ortega, Stanford, CA 94305, USA. [4] University of Washington, School of Marine and Environmental Affairs, 3707 Brooklyn Ave, Seattle, WA 98105, USA. [5] Monterey Bay Aquarium Research Institute, 7700 Sandholdt Road, Moss Landing, CA 95039, USA. [6] Florida Fish and Wildlife Research Conservation—Fish and Wildlife Research Institute, 100 8th Avenue SE, St. Petersburg, FL 33701, USA. *email: anni.djurhuus@gmail.com; closek@gmail.com; mya@usf.edu

Our ability to track changes in marine ecosystems is hampered by an inability to simultaneously assess the abundance and distribution of diverse organisms across multiple trophic levels. Traditional marine biodiversity assessments frequently focus on macro-organisms and are conducted via visual and manual methods such as diver surveys, trawling, acoustic techniques or aerial imagery[1]. More recently, global and local surveys using genetic methods have been conducted to explore single-celled communities, unveiling the vast diversity of marine microbes[2,3] and small eukaryotes[4]. However, exploring the diversity and distribution of a broad swath of prokaryotic and eukaryotic life forms in the world's oceans remains a major challenge, especially when examining interactions and co-occurrences of taxa and trophic levels (i.e., microbes to mammals) on the same temporal and spatial scales.

Amplicon sequencing of DNA derived from environmental samples (often termed environmental DNA (eDNA) meta-barcoding) is well established in the field of microbial ecology and has more recently been applied to multicellular organisms[5–7]. Examination of eDNA genetic sequences has emerged as a method to infer co-occurrence patterns of organisms within an ecosystem, across all domains of life simultaneously[5,8,9], although this is not common practice yet[7]. So far, these methods have been used to provide essential information about the spatial distribution and temporal variability in single-celled organisms spanning large areas in the world's oceans[2,4,10–13]. However, a major goal is to holistically explore complete biological communities, surveying the diversity and distribution of single- and multi-cellular organisms alongside environmental parameters to reveal the interactions across community members and predict how whole communities respond to environmental change.

Here, we analyse a time-series of eDNA metabarcoding data to assess changes in the proportional abundance of taxa in a marine setting, and to analyse linkages of organisms ranging from microorganisms to mammals. This coordinated survey of entire biotic communities using high-throughput sequencing of multiple conserved genetic markers from eDNA elucidated relationships between community dynamics and environmental properties. This approach allowed us to test hypotheses about the predicted association between the richness of taxa and environmental variables, including sea surface temperature, chlorophyll *a*, and other key environmental and biotic variables. Through network analysis, we found that groups of co-occurring organisms spanning different trophic levels were directly correlated to changes in environmental parameters, providing insights into the underlying response of whole communities to the environment and highlighting co-occurrences and potential trophic interactions.

## Results

**Temporal community structure.** We collected seawater samples approximately bimonthly for 18 months ($n = 8$ time points, April 2015–December 2016) from a long-term monitoring station in Monterey Bay National Marine Sanctuary (MBNMS), CA, USA (Supplementary Fig. 1). Environmental variables (including water column temperature, salinity, dissolved oxygen, chlorophyll *a* and nitrate) were measured in situ or from seawater samples at all sampling time points (see Methods). eDNA was concentrated from seawater on membrane filters. Four genetic loci (16S ribosomal RNA (rRNA), 18S rRNA, cytochrome *c* oxidase I (COI), and 12S rRNA) were amplified and sequenced (see Methods and Supplementary Table 1 for details on laboratory and bioinformatic analysis). Over $10^8$ sequences were recovered after quality control (see Methods, e.g., removal of negatives, Supplementary Fig. 2), resulting in the identification of 663 taxonomic groups

(grouped and analysed at a Family or higher taxonomic level for the purpose of this study, see Methods). The taxa identified employed saprotrophic, autotrophic, mixotrophic and hetero-trophic trophic strategies. Because amplification bias obscures the relationship between organismal abundance and amplicon abundance[14] (see Methods), we created indices of abundance for each annotated taxon, scaled from zero to one. This method assumes that amplification bias arises from template–primer interaction, and that for any given taxon-primer pair, this interaction is constant across samples, allowing us to infer relative changes in abundance between different taxa[15]. When a taxon was detected with multiple genetic loci, we averaged these indices to create an ensemble index for that taxon. We then measured pairwise correlations (Kendall's tau) for taxon eDNA indices to detect clusters of taxa with simultaneous changes in amplicon-index abundance. We permutated a null model to correct our false-discovery rate for multiple comparisons (Supplementary Fig. 3 corrected significance threshold: tau = 0.70). Given our focus on marine ecosystems, all instances of terrestrial organisms that appeared on the taxon list were removed, although we recognise that some terrestrial taxa have aquatic life cycle stages and may be ecologically important. After removing terrestrial taxa, 348 unique marine taxa remained, 274 (78%) of which were agglomerated at the Family level, and otherwise pooled at a higher taxonomic level (Order, Phylum, etc., see Methods).

To evaluate the changes in proportional abundance of the community with time, we used weighted gene correlation network analysis (WGCNA)[16] of the amplicon abundance index of each taxon. This identified clusters of taxa with similar trends over time (Fig. 1a, b). The WGCNA builds a network with taxa (nodes) linked (edges) to other taxa with similar trends[17] (Supplementary Fig. 4, see Methods). Interconnected subnetworks (communities) are identified from the main network, representing collections of taxa from all domains of life and trophic levels with strongly correlated shifts in proportional abundance over time. The correlated changes may stem from direct or indirect trophic interactions or from coincident responses to environmental factors[18].

We identified six subnetworks (based on the network analysis for the whole dataset, see Methods) that represent different communities and their changes with time (Fig. 1a, b and Supplementary Fig. 4). Although representative taxa from each subnetwork occurred at most time points, the subnetworks reflected seasonal changes in the MBNMS community. The eDNA observations on richness between spring/summer taxa (orange, yellow, green subnetworks) were visibly distinguishable from autumn/winter taxa (grey, blue, black) (Fig. 1a, b). All subnetworks were correlated against environmental variables; for example, the seasonality in taxon richness (Fig. 1b) reflects significant correlations with lower chlorophyll *a* levels during winter (grey subnetwork, $r = -0.76$, Spearman correlation *p* value = 0.03) and higher sea surface temperatures during autumn (blue subnetwork, $r = 0.74$, Spearman correlation *p* value = 0.04) (Supplementary Figs. 5 and 6).

The absolute highest and lowest observed taxon richness (i.e., total number of taxa) occurred in December 2015 and December 2016, respectively, when the grey network was dominant. This expansion/contraction of diversity within a single subnetwork provides some insight as to how communities might respond to interannual differences such as the above average sea surface temperatures and changes in physical circulation in December 2015 due to a strong El Niño event (Supplementary Fig. 7), which has been previously observed in eDNA studies[19]. Other studies in this region have also reported changes in community structure relative to other years during El Niño events, including anomalous species richness, when taxa from warmer regions to

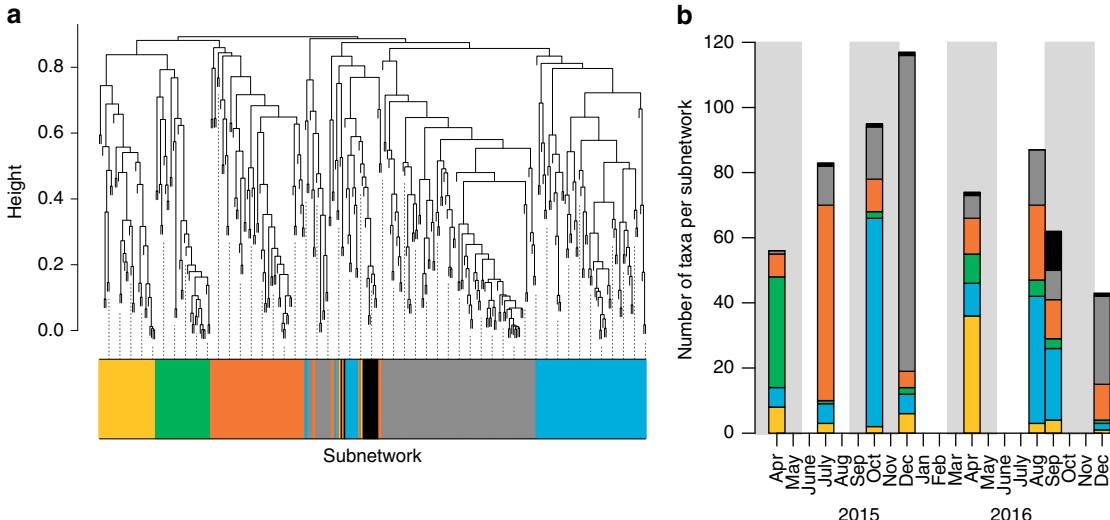

**Fig. 1 Weighted gene correlation network analysis (WGCNA) of eDNA data correlated with environmental variables. a** Dendrogram based on clustering of changes in abundance indices of all taxa using Kendall's tau correlation coefficient (see Methods). The colours correspond to different subnetworks. **b** Observed richness of taxa from each subnetwork, from Fig. 1a, over the sampling time points showing the highest accumulative richness in December 2015. The shaded areas represent the spring and autumn seasons and white represents the summer and winter seasons.

the south and/or offshore, move north toward the central California coast[20].

**Composition and temporal behaviour of subnetworks**. We expect taxa with strong trophic interactions to have highly correlated patterns of proportional amplicon abundance over time[20]. To evaluate these trophic interactions, we assume that correlations among taxa more than one trophic level apart are not direct, with the exception of ecto- and endosymbiotic organisms. We observe a few likely indirect connections among taxa, and a variety of direct connections that suggest trophic interactions.

The dominant spring subnetworks of 2015 and 2016 (green and yellow, respectively) contained most of the centric diatom taxa (30% and 41%, respectively) in the highest proportional abundances (Supplementary Fig. 8). These subnetworks were associated with early seasonal upwelling and had the least diverse autotrophic communities, albeit the most centric diatoms, which are known to cause spring blooms in the region[21]. Typically in MBNMS pennate diatoms appear after the centric diatoms[21]; however, we found pennate diatom taxon richness distributed equally across all subnetworks (and thus time), with greater proportional abundances during spring and winter 2015 (Supplementary Fig. 8). The summer/autumn subnetworks (orange and blue) contained a large fraction of all dinoflagellate taxa (26% and 20.5%, respectively). This is expected for the latter part of the upwelling season and the transition to non-upwelling conditions[21]. However, during our sampling period, the grey subnetwork (dominant in December 2015 and December 2016) contained the highest fraction of dinoflagellates (35%), which is consistent with minimal seasonal upwelling at that time and was compounded in late 2015 by El Niño conditions, when winter sea surface temperatures can be higher than average and more prone to stratification[22]. In addition to high dinoflagellate abundance, the proportional copepod abundance was also highest during winter 2015 (Supplementary Fig. 8), representing 38% of the total number of taxa at that time (grey subnetwork).

The autumn (blue) and winter (grey) subnetworks showed the strongest correlations with environmental parameters. We focus on these subnetworks to illustrate the relationship between environmental conditions and putative ecological responses (Supplementary Figs. 5 and 9). The dominant subnetwork in

autumn (blue) comprised a total of 81 taxa that were more common when sea surface temperature was >14 °C (Fig. 2a–c and Supplementary Fig. 7). The blue subnetwork contained phytoplankton taxa that can cause harmful algal blooms (e.g., the toxic algal families Gonyaulacaceae and Prorocentraceae), which have been observed previously as a response to positive temperature anomalies in the MBNMS[23,24]. Balaenopteridae (sequences identified as humpback whale; *Megaptera novaeangliae*), commonly observed in MBNMS, was a top predator and the most highly connected taxon (most central node, see Methods) within the blue subnetwork (Fig. 2a). Krill (Family Euphausiidae), a common prey of humpback whales[25], was classified within the same subnetwork, however its correlation to Balaenopteridae was not strong (Fig. 2, Supplementary Figs. 10 and 11).

The dominant winter (grey) subnetwork was comprised of 114 taxa and positively associated with low chlorophyll *a* (Fig. 2d–f). Within this subnetwork we observed likely direct connections among taxa with simultaneously high amplicon-abundance indices, i.e., Otariidae (sequences identified as California sea lion; *Zalophus californianus*; top predator) and Carangidae (sequences identified Pacific jack mackerel; *Trachurus symmetricus*; next highest trophic level). California sea lions prey upon jacks including the Pacific jack mackerel, which occur in MBNMS[26,27]. Jacks in turn often feed upon planktonic copepods[28], and indeed, copepods from the grey network (especially Metridinidae) were strongly correlated to jacks ($r = 0.80$, Spearman correlation $p$ value < 0.05). The grey subnetwork contained several taxa with parasitic life histories, including the Family Syndiniaceae (dinoflagellates), which are parasitic to a broad range of hosts, including crustacea, radiolaria and fish, which were also present in the grey subnetwork (Supplementary Figs. 8, 10, and 11). Most radiolarians (90%), including all celestine (Acantharea) forms, were associated with the winter 2015 (grey) subnetwork. Siliceous (Polycystinea) and celestine radiolarians can graze on other plankton and some taxa have dinoflagellate, hapytophyte or prymnesiophyte endosymbionts, which may allow members of this group to dominate during periods when diatoms are less prevalent, as seen within the winter network. Therefore, this approach of eDNA metabarcoding combined with network analysis deciphers subnetworks of co-occurring taxa that we can use to discover putative ecological interactions.

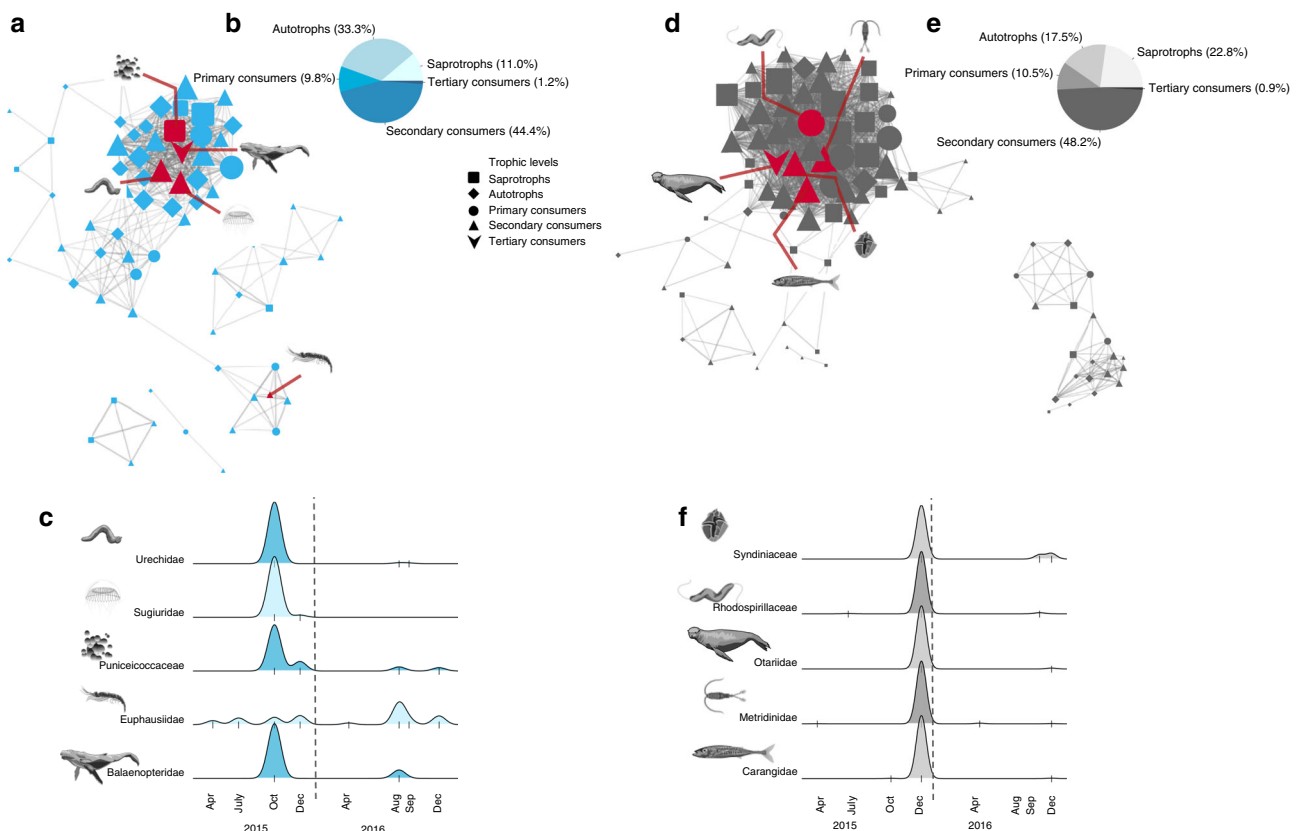

**Fig. 2 Network visualisation and amplicon-index abundance of selected taxa over time.** Network visualisation for the autumn 2015 and 2016 (blue) subnetwork (**a–c**) and the December 2015 (grey) subnetwork (**d–f**). The blue (**a**) and grey (**d**) subnetworks are visualised with nodes (taxa) and edges (correlations) representing the connections between the individual taxa. To best visualise the subnetworks, only taxa with a network connection (edge weight) above a threshold of 0.2 are shown (i.e., low correlations were removed, see Methods for edge calculations). The different trophic levels detected within the subnetworks are represented by different symbols, see legend. These panels illustrate the complexity of the co-occurring taxa in the blue and grey subnetworks, upon which the taxa representations of panels (**c**) and (**f**) are chosen. The distribution of the trophic levels within each subnetwork is represented in (**b**) and (**e**). The size of the taxon node symbol is relative to the number of edge connections that an individual taxon has within the subnetwork and the width of the edges represents the weight of the correlation between two given nodes. The red lines within each subnetwork point to highly correlated taxa ($r > 0.9$) or highly connected taxa (within the top 10%) within the subnetworks, highlighted in (**c**) and (**f**). The scaled abundances of these taxa are presented in (**c**) and (**f**) during the course of our sampling period. The tick marks in the x-axis of plots (**c**) and (**f**) represent the times of sampling when we have observations of these taxa, the height of the y-axis is relative to the number of observations per taxon and does not represent absolute abundance values or biomass. The vertical dashed lines represent January 1, 2016.

**Correspondence of networks with environmental shifts**. The blue (autumn) and grey (winter) subnetworks were associated with high temperature and low chlorophyll *a* concentrations, respectively, indicating coherent shifts in biological communities associated with changes in environmental conditions. Within these subnetworks, amplicon abundance indices (see Methods) of constituent taxa were predictably correlated with the same covariates (Fig. 3a, e.g., most of the taxa from the winter subnetwork, which was negatively correlated with chlorophyll *a*, were also negatively correlated with dissolved oxygen).

In the autumn subnetwork, most of the taxa with the strongest positive correlations with temperature are notably also the taxa with the highest intra-subnetwork connectivity (see Methods) and the highest subnetwork membership (i.e., highest sum of subnetwork correlations to other taxa (or edge weights)) (Supplementary Table 3). For the autumn and winter subnetworks, the correlation (Spearman *r* correlation coefficient) of individual taxa to the environmental variable (temperature and chlorophyll *a*, respectively), is significantly correlated with the degree of connectedness of those taxa (i.e., the more connected a taxa is, the more significantly it is correlated with the environmental variable, Fig. 3b, c).

It has recently been shown that the degree of centrality can be used to identify keystone taxa with 85% accuracy in microbial studies[29], although this has been met with some criticism[30,31]. This predictive capability has not yet been empirically tested in studies across trophic levels from microorganisms to mammals. Amongst the most interconnected taxa (top 10%) and taxa significantly correlated with temperature or chlorophyll *a* (Fig. 3b, c, above dashed red line), we found representatives of all trophic levels and across all domains of life. From the autumn (blue) subnetwork, examples of top representatives of different trophic groups include the Urechidae[32] (polychaete), Rathkeidae (protist), and Planctomycetales (bacteria) (Supplementary Table 3). These taxa represent a secondary consumer, primary consumer and a saprotroph, respectively.

Within the winter (grey) subnetwork, examples of representatives from the top 10% connected taxa are the Polycystinea (radiolarian), Carangidae (jacks) and Thalassoarchaea (archaea), representing a primary consumer, secondary consumer and a mixotroph, respectively. The tertiary consumers in each subnetwork (humpback whales and California sea lions) were also among the 10% most highly connected taxa at ranks one and four, respectively (Fig. 3 and Supplementary Table 3). The single

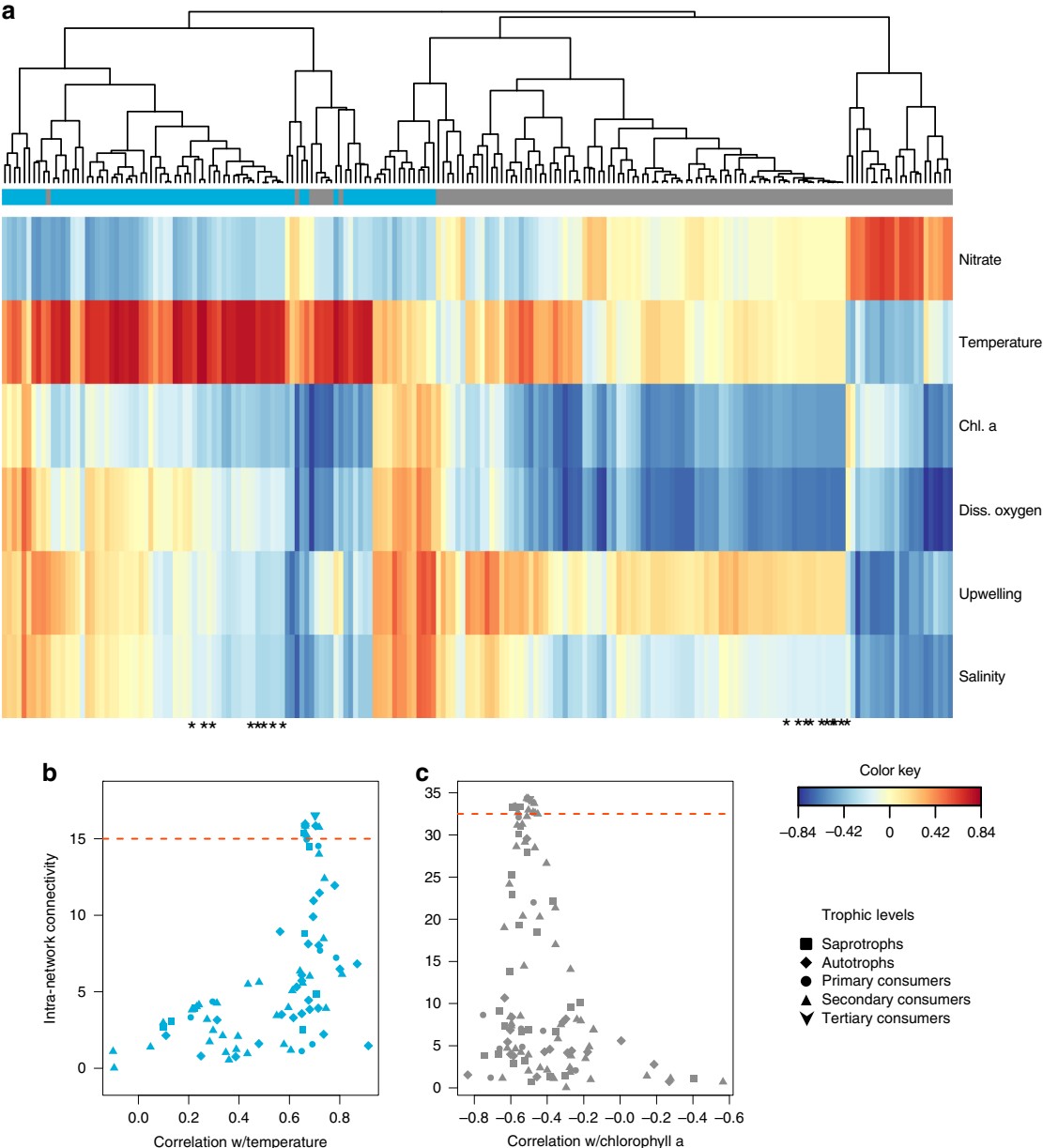

**Fig. 3 Partial least square analysis between taxa and environmental variables, and plot of taxon connectedness and correlation to the environment. a** Partial least square analysis plot showing the correlation of all taxa (columns), clustered by Kendall's tau correlation coefficient, in the grey (winter) and blue (autumn) subnetworks (leaf colour) with environmental variables (rows). The heatmap is coloured by the value of the correlation coefficient, see colour key. The asterisks represent the top 10% most connected taxa in each subnetwork (above the red horizontal line in (**b**) and (**c**)). **b, c** The correlation of the connectedness or centrality of each taxon within the blue (**b**) and the grey (**c**) subnetworks correlated to temperature and chlorophyll *a*, respectively. Symbols correspond to the trophic levels shown in the legend.

central hub taxa with the highest connectivity to other taxa within the respective subnetworks were Balaenopteridae (humpback whales) for the blue subnetwork and Rhodospirillaceae (proteobacteria) for the grey subnetwork. These taxa were connected to a total of 24 (29%) and 48 (42%) taxa within their respective subnetworks. The difference of number of connected taxa between the top ranked taxa in the two subnetworks is due to the difference in numbers of taxa within that particular subnetwork (blue: $n = 81$, grey: $n = 114$).

Having a quantifiable threshold for consistent identification and validation of indicator or keystone taxa is important[33], and requires experimental evidence showing the impact of the respective taxa on community function and composition. Network interactions alone can show positive associations between taxa that are unlikely to interact directly (e.g., Urechidae (a sediment dweller) and humpback whales), and it is important to keep in mind that such correlations are manifestly not causation[34]. However, such networks of putative interactions highlight fluctuations of taxa with their environment that may have previously escaped notice, and thus may be powerful ways of identifying novel organisms and community assemblages that could indicate environmental changes and pioneer conservation strategies, which would especially benefit from these types of analyses at the species level (see Supplementary Fig. 11).

## Discussion

In light of global climate change and increasing uses of ocean spaces and resources, monitoring marine biodiversity has

emerged as a fundamental requirement to define conservation priorities[35]. Most marine time-series studies have focused on determining variations in a limited group of organisms (bacteria, phytoplankton, zooplankton, fish, mammals, etc.), often relative to environmental conditions and more rarely exploring biotic interactions between them. Comprehensive ecosystem studies are hampered by the lack of practical methods and expertise needed to quantify biodiversity. The development of eDNA methods provides a unique lens into marine biodiversity by enabling the simultaneous examination of organisms across multiple trophic levels and domains of life that can provide critical information about the complex biotic interactions related to ecosystem change.

Assessing the temporal variability of complex biological communities is challenging because individual taxa respond differently to changing environmental conditions. Our results confirm that groups of marine organisms can be effectively clustered into communities (subnetworks) that fluctuate in composition and abundance over time. Disentangling the total biodiversity of a location can help identify previously unknown taxa that could serve as keystone species, which in turn may be used as indicators of change due to external environmental or biotic factors (i.e., temperature, pH, nutrient availability or food web interaction). Applying eDNA metabarcoding to monitor biodiversity in longer time-series will increase the power and validity of this technique for predicting ecosystem response to climate change and help inform conservation strategies. Biodiversity fluctuations with time would ideally be observed on a species level, albeit the lower the taxonomy assignment, the lower the confidence in that assignment becomes. This emphasises the importance for database expansion and development for eDNA taxonomic assignments.

Using eDNA can also help identify cryptic taxa and detect taxa that are difficult to identify (e.g., the Family Paralichthyidae (benthic flat fish), detected in this study), although it does not allow us to identify stages of life history and could include sequences from dead or decaying biological matter. Amplification of DNA is subject to substantial primer biases and is almost certainly affected by inhibitors and variable abundances of template DNA; these complications likely affect the number of taxa detected with these methods, and consequently, we note that no method—including this one—identifies all taxa in an ecosystem. However, eDNA analysis can combine observations across orders of magnitudes of organismal body size (i.e., microbes to mammals), across many trophic levels, and across vastly different domains of life. This property allows us to explore target species for conservation that could be secondarily combined with traditional methods (e.g., visual identification for abundance measures) for a more thorough survey of specific organisms or communities of interest. These analyses can predict taxon interdependencies that should be further investigated through hypothesis-based research and experiments.

We illustrate the utility of eDNA metabarcoding to create hypotheses that expand upon well-documented predator–prey interactions (e.g., baleen whales and krill in the blue subnetwork) by including previously unexamined trophic connections, such as the co-occurrence of specific primary producers and microbial groups. Within the grey network, the co-occurrence of the taxa Otariidae (sea lions), Carangidae (jacks) and Metridinidae (copepods) are putative predator–prey relationships. Pacific jack mackerel was a particularly important component of the California sea lion diet in 2015[27], whether their co-occurrence in our study is driven by direct predator–prey relationships or is instead a common response to an external driver is unclear. Our network analyses also indicated the co-occurrence of several additional taxa within this subnetwork, including Pelagibacteraceae (bacteria), Rhodospirillaceae (bacteria) and two parasitic groups, the

Ellobiopsidae (protists) and Syndiniaceae (dinoflagellates) that support the 'predator–prey' co-occurrence hypothesis. The Ellobiopsidae and Syndiniaceae are known parasites of copepods, radiolarians, crustaceans and some fish, which lends support to the hypothesis that linkage of taxa within a given subnetwork reflects ecological interactions (see Supplementary Fig. 11, for an example of species level networks). Furthermore, taxa in different subnetworks may nevertheless interact. For example, despite the grouping of anchovies within the green subnetwork, they have known trophic interactions to baleen whales, which are in the blue subnetwork. This is simply an example of a higher correlation and co-occurrence of anchovies with their potential prey (copepods) within the green subnetwork[36] and further highlights the importance of taxa succession in monitoring predator–prey interactions over time.

Although network connectivity does not necessarily indicate interaction, a more connected taxon is more likely to present similar patterns of abundance, occurring at the same times as other taxa from that subnetwork (i.e., it is highly correlated with more taxa within that network than a less connected taxon). The Balaenopteridae (humpback whales) and Rhodospirillaceae (proteobacteria) are the most connected taxa within the blue and grey subnetworks, respectively. Given that the most connected taxa (see Methods) are highly sensitive to environmental variables, we hypothesise that these taxa may represent potential indicators of different ecosystem states[34], similarly to keystone species predicted from microbial networks by Banerjee et al. (2018)[31,34,37]. These taxa could be used as early-warning indicators of regime shifts through entire biomes following ecosystem-scale events, such as El Niño cycles, upwelling, or anthropogenic disturbances, which would most likely be detected by eDNA over a longer time period[19]. The more highly connected taxa within different subnetworks at a given site can potentially be considered Essential Ocean Variables[38]. Such variables are required by biodiversity assessments, remote monitoring tools with built-in biological sensors, and ecosystem models. eDNA provides one potential means for identifying and monitoring indicator taxa, and the incorporation of these data into long-term monitoring programmes, such as the marine biodiversity observation network (MBON)[39,40], can help address national and international needs for practical measures of biodiversity.

We conclude that when applied to an ecosystem over time, surveys using eDNA analysis can yield data-based biological indicators of ecosystem change. Increasing such datasets can improve forecasting of biodiversity shifts that may result from environmental changes that occur across varied time scales and locations[19]. The essential biological observations provided by this technique will aid future efforts for proactive conservation of life in the world's oceans.

## Methods

**Sample collection and laboratory methods.** Sampling was carried out on the R/V *Rachel Carson* and *Western Flyer* bimonthly at the permanent Monterey Bay (MB) time series station C1 (36.797°N, 121.847°W) (Supplementary Fig. 1). Seawater samples for eDNA were collected using Niskin bottles on a CTD rosette at approximately 0–1 m depth. At each sampling point, a single 1 l water sample was filtered onto a 0.22 μm pore size polyvinylidene difluoride membrane filter (Millipore, USA). All filters were flash frozen in liquid nitrogen and preserved at −80 °C until further analysis. The team sacrificed biological replication for more distinct samples and a higher sequencing depth per sample based on previous studies resulting in minimal differences between replicates[8,9,41].

DNA extraction was performed on all membrane filters using the Qiagen DNeasy Blood and Tissue Kit with modifications according to Djurhuus et al. and Walz et al.[41,42]. Subsequently, all samples were metabarcoded for the 16S rRNA[43], 18S rRNA[44], COI[45] and 12S rRNA[46] genes (see Supplementary data for sample barcodes). Polymerase chain reaction (PCR) reactions, artificial communities (positive) and a non-template control (negatives) together with DNA extraction blanks (blanks) were run in triplicates (see Supplementary Fig. 2). The replicate PCR products were pooled for each sample by genetic marker and run through an

agarose gel to confirm the presence of target bands, clean negatives and the absence of non-specific amplification. PCR products were purified and size selected using the Agencourt AMPure XP bead system (Beckman Coulter, USA). A second agarose gel was run to confirm primer removal and retention of target amplicons after purification. Purified products were then quantified using the Quant-It Picogreen dsDNA Assay (Life Technologies) on an fmax Molecular Devices Fluorometer with SoftMaxPro v1.3.1 (16S rRNA, 18S rRNA and COI genes) or using a Qubit dsDNA HS kit (12S rRNA gene). Equimolar pools were constructed and quantified to confirm pool concentration prior to library preparation. One library was constructed from the pooled product for each genetic locus using the KAPA HyperPrep and Library Quantification kits following manufacturer's protocol. Libraries were loaded on a standard MiSeq v2 flow cell and one sequencing run per genetic locus was performed in a $2 \times 250$ bp paired end format using a v2 500-cycle MiSeq reagent cartridge. For the 16S rRNA, 18S rRNA and COI genes the MiSeq runs were performed with a 10% PhiX174 spike in, while for 12S rRNA 20% PhiX174 was added. Custom sequencing primers were added to appropriate wells of the reagent cartridge. Base calling was done by Illumina Real Time Analysis (RTA) v1.18.54 and the output of RTA was demultiplexed and converted to FastQ format with Illumina Bcl2fastq v2.18.0.

**Bioinformatics**. Resulting sequences from the four libraries (16S rRNA, 18S rRNA, COI, and 12S rRNA) were processed through a modified version of the banzai pipeline Unix shell script[47]. Paired-end reads were assembled and filtered with PEAR[48]. Homopolymers were removed with grep and awk commands. Samples were concatenated, and tags were removed. Primers were removed with cutadapt (Martin, EMBnet) and singletons were removed. Operational taxonomic units (OTUs) were clustered with Swarm[49]. Chimeras were removed with VSEARCH v1.8.0.

Taxonomic annotations for 16S rRNA were performed with GreenGenes 13.5 downloaded on December 17, 2016. Taxonomic annotations for 18S rRNA, COI and 12S rRNA were performed with the GenBank nr BLASTN database that was downloaded from NCBI on September 20, 2017. The max target sequence within the BLAST algorithm was interpreted according to Shah et al.[50]. Annotations with >80% identities were retained (Supplementary Table 1). These annotations were then interpreted through MEGAN6, which only considered hits that had a bitscore of greater than 100 and were within the top 2% highest scoring hits per contig. The most recent common ancestors of these hits were subsequently determined.

**Occupancy modelling of sequencing data**. For each individual-locus dataset, the first step of decontamination was to determine the OTUs likely to be truly present in the dataset vs. those likely to be false-positive discoveries (e.g., artefacts of PCR or sequencing), see Supplementary Fig. 2. We used a site-occupancy model to estimate the probability of OTU occurrence[51,52], using multiple PCR replicates of each environmental sample as independent draws from a common binomial distribution. We eliminated any OTU with <80% estimated probability of occurrence (a break point in the observed distribution of occupancy probabilities) from the dataset.

**Decontamination of sequencing data**. Further decontamination followed the procedure described in ref.[53]. This entailed (1) subtracting the most-likely OTU-specific proportional contribution of contamination from each OTU in the field samples, blanks and negatives to minimise the effect of potential cross-contamination among samples due to tag-jumping[54] or similar effects; and (2) dropping samples that had highly dissimilar PCR replicates (Bray–Curtis dissimilarities >0.49, which were outside of the 95% confidence interval given the best-fit model of the observed among-replicate dissimilarities).

**Abundance indexing of sequencing data**. After decontamination we grouped all OTUs by their Family annotation or, if an OTU could not be annotated to Family, to Order or Class. In this manuscript we refer to a taxon as an individual Family, Order or Class, using the most specific taxonomic resolution available at or above Family level. One taxon might represent several OTUs, species or genera, but due to the sheer number of OTUs, species and genera, all analyses were done at higher classifications to simplify the results. However, the analysis could potentially be done at a lower taxonomic level, if desired, understanding that species-level annotations may not be reliable for some taxonomic groups and, for some species complexes, species cannot always confidently be discerned with a targeted gene loci (see Supplementary Fig. 11).

Metabarcoding datasets are a product of many analytical steps including from sampling technique, DNA extraction method, amplification with particular primers, sequencing depth and technique, quality control and bioinformatic processing. Each of these steps might influence the number of reads assigned to a particular taxon in a given sequencing run. Comprehensively understanding the sources of this bias is a significant undertaking[55]. However, it is likely that PCR and primer bias is by far the largest source of variance in the eDNA process: out of the same environmental extract, different primer pairs produce completely different sets of taxa for analysis[9,14,56]. As such, combining information from eDNA analyses that use different primer sets is an important opportunity to develop a more comprehensive sampling of taxa in a given environment. To combine

information recovered from different loci, we standardised each dataset by determining the proportion of reads assigned to each taxon within each water sample and then dividing these by the largest observed proportion for each taxon to create a taxon-specific index of eDNA reads that varied (across samples) between zero and one (1) into an eDNA index, see Kelly et al.[15].

$$\text{eDNA}_{ij} = \frac{\frac{Y_{ij}}{\sum_i Y_i}}{\max_j \left( \frac{Y_{ij}}{\sum_i Y_i} \right)}. \tag{1}$$

This normalisation reflects the intuition that raw read counts, standing alone, do not provide reliable information about the abundance of taxa present—rather they are information about the interaction of a particular primer set with a particular taxon template. These indices of abundance are calculated only within-taxon, within-locus. Moreover, because we can treat individual loci as effectively independent, we then create an ensemble index of abundance—for each taxon at each time point—by taking the mean of the indices for that taxon across different loci. If a locus does not amplify a taxon, it provides no information about that taxon; consequently, we include in the ensemble index only non-zero indices for each taxon. The result is an ensemble index that appears to behave sensibly with respect to individual-locus observations, and that allows us to track changes in abundance of eDNA for many individual taxa simultaneously.

**Correlation and correction for multiple comparisons**. This is a short-time-series dataset, with hundreds of taxa varying in abundance over eight time points. Consequently, (1) the number of pairwise comparisons is far greater than the number of data points within a single comparison, and (2) individual comparisons of eight rank-abundance values (some of which are zero) result in many ties. We used Kendall's tau as our measure of rank correlation to best handle these ties. We chose Kendall's tau because it is robust to outliers (i.e., the difference in rank between the two highest points is equal to that between the two lowest). However, there is no general null-probability distribution for values of tau, making it difficult to precisely distinguish observed correlations from those expected by chance alone. Doing so is particularly critical when correcting for multiple comparisons: because we have tens of thousands of pairwise comparisons, many will have large values even under the null distribution. We therefore generated a null distribution and determined statistical significance for correlations as follows:

1. We randomly permuted our dataset to generate 100 null versions, and carried out pairwise Kendall correlations for all taxa within each of those permuted versions, creating a null distribution of Kendall's tau with a total of $6.5 \times 10^4$ pairwise comparisons.
2. For each value of tau, we then calculated the proportion of null datasets containing that value or greater, resulting in a probability of seeing a given value of tau by chance alone. This was effectively the probability density distribution for tau, given our data.
3. We then compared our observed correlations to this null distribution to derive a $p$ value for each of our observed data points.
4. We then adjusted this $p$ value for multiple comparisons—our dataset required $6.5 \times 10^4$ pairwise comparisons—using the mean Benjamani–Hochberg false-discovery rate adjustment (in this application, the adjustment is also equal to the Bonferroni correction), and considered all correlations having an adjusted $p$ value smaller than 0.05 to be significant. Our critical value of tau was 0.7; we counted all values above this threshold as significant after $p$ value correction.

**Trophic assignments of taxa**. We further characterised taxonomic annotations by assigning a trophic assignment and additional life history characteristics. Trophic assignments were assigned to a scale of 0–5 (Archaea, Bacteria, and fungi, primary producers, primary consumers, secondary consumers and tertiary consumer). Bacteria were characterised as mixotrophic if the taxon uses sunlight-coupled oxidation in addition to heterotrophy. We used the National Centre for Biotechnology (NCBI), World Register of Marine Organisms (WoRMS), Encyclopaedia of Life (EOL), Integrated Taxonomic Information Systems (ITIS), taxa-specific databases, Wikipedia and peer-reviewed papers served as sources to verify trophic assignments and assign a group classification (e.g., copepod and parasitic). Taxa trophic and group assignments were assigned again by a second person to verify the reliability of assignment and finally confirmed with a third assignment check.

**Weighted correlation network analysis**. To determine the weighted correlation we calculated a correlation matrix containing all pairwise Kendall tau correlations between all taxa across all samples[17]. We define correlation networks as undirected, weighted taxa networks. The nodes of the network correspond to taxa, and edges between taxa are determined by the pairwise Kendall's tau correlations between the amplicon-index abundance of the taxa (Supplementary Fig. 3). By raising the absolute value of the Kendall's tau correlation to a power $\beta$, the weighted correlation network construction emphasises large correlations at the expense of low correlations. To calculate the weighted correlations we used the formula, $aij = |\text{cor}$

$(xi, xj)|\beta$. This represents the adjacency of a weighted taxa network. We used the scale free topology criterion to choose the soft threshold $\beta = 22$[16].

To organise taxa into subnetworks, we used the topological overlap measure as a robust measure of interconnectedness in a hierarchical cluster analysis[16,17]. The result is a list of taxa (nodes) and their connectedness to each other (weight) based on Kendall's tau correlation raised to the power of 22 divided by clustering patterns, i.e., taxa that showed similar patterns over time were subdivided into the same subnetworks with varying degrees of correlation/similarity. We used the Dynamic Tree Cut function to divide the clusters into subnetworks, see: https://peterlangfelder.com/2018/12/30/why-wgcna-modules-dont-always-agree-with-the-dendrogram/.

All subnetworks were examined for correlation to all environmental variables using a Pearson correlation (Supplementary Fig. 5). The intra-network connectivity was calculated from the sum of edge weights (degree of correlation) the individual taxa had.

**Sparse partial least squares analysis.** In order to directly associate specific taxa to environmental variables (Fig. 3), we used sparse partial least square (sPLS) analysis from the R package mixOmics[57,58]. We applied the sPLS in regression mode, which will model a causal relationship between the lineages and the environmental traits, that is, PLS will predict environmental traits (e.g., temperature) from lineage abundances. This approach enabled us to identify strong correlations between certain taxa and environmental variables without taking into account the global structure of the planktonic community.

**Reporting summary.** Further information on research design is available in the Nature Research Reporting Summary linked to this article.

## Data availability

The data have been deposited with links to BioProject accession number PRJNA433203 in the NCBI BioProject database (https://www.ncbi.nlm.nih.gov/bioproject/). The source data underlying Figs. 1–3 and Supplementary Figs. 2, 4, 7, 8 and 10 are provided as a Source Data file. The map image in Supplementary Fig. 1 was created using the m_map mapping toolbox for MATLAB® commercial software. The coastline shown in the map is based on data from the Global Self-consistent, Hierarchical, High-resolution Geography Database (GSHHG): https://www.ngdc.noaa.gov/mgg/shorelines/gshhs.html.

## Code availability

All scripts for analysis are available on github: https://github.com/marinebon/eDNA_microbes_whales.

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

## Acknowledgements

This work is a contribution to the Marine Biodiversity Observation Network (MBON). The MBON project was supported by NASA grant NNX14AP62A 'National Marine Sanctuaries as Sentinel Sites for a Demonstration Marine Biodiversity Observation Network (MBON)' funded under the National Ocean Partnership Programme (NOPP RFP NOAA-NOS-IOOS-2014-2003803 in partnership between NOAA, BOEM and NASA), and the U.S. Integrated Ocean Observing System (IOOS) Programme Office. The authors would like to acknowledge the crew on board the R/V Western Flyer and R/V Rachel Carlson. We acknowledge Paul Matson, Emily Jacobs-Palmer and Peter Langfelder for feedback on the analysis of this work.

## Author contributions

A.D., C.J.C., R.P.K., K.J.P., R.P.M., K.R.W., F.P.C., A.B.B. and M.B. conceived the study. A.D., C.J.C. and R.P.K. wrote the paper. D.O., E.M. and F.E.M.K. analysed oceanographic data. A.D., C.J.C., K.J.P., H.A.S., K.R.W., E.O. and K.H. performed the genomic lab analyses and A.D., C.J.C., K.J.P., R.P.M., R.P.K. and K.R.W. performed the bioinformatics. C.J.C., E.O., K.H., K.J.P., M.B., E.A.A. and A.B.B. assigned trophic levels to all taxa. A.D. and R.P.K. performed the statistical analyses and A.D. produced and analysed the networks. A.D. produced all figures and E.O. made the figure illustrations. All other contributed to the interpretation of the networks and discussed and commented on the paper.

## Competing interests

The authors declare no competing interests.
