## [Peer Review File · Nature Communications]

Reviewers' Comments:

Reviewer #1:

Remarks to the Author:

The authors describe an environmental DNA time-series from Monterey Bay that provides insight into co-occurrence of organisms ranging from the microscopic to the large and multicellular. The authors perform a range of analyses but focus most of the paper on the WGCNA network clustering analysis.

Overall the paper is well-written and easy to follow. The figures are informative and well-crafted. A major issue with correlation networks is that interactions are often assumed from correlations, but the authors are cautious not to fall into this trap. The results are described robustly and accurately. In terms of methods, it is very difficult and tedious to merge datasets from different amplicon sequencing targets, and the authors have presented a large array of quality-checks to verify that their results carry biological meaning. There will always be some vagaries when it comes to OTU clustering and taxonomic assignments, but given current limitations the authors have done a good job.

This work fits the criteria of Nature Communications and would be of interest to a broad readership spanning marine science, conservation biology, ecology, and microbial ecology.

A few points:

-One issue I am wondering about. The abundance of sequences in a given sample will be related to abundance for microbial taxa, but not for metazoans. One line 301 the authors state that baleen whales and Rhodospirillaceae are highly connected, and therefore likely present "in a similar relative magnitude". In my mind this must mean that the two organisms co-occur in that sample, but nothing more. If one finds a large number of Rhodospirillaceae sequences in a library one can usually assume (not always accurately) that they are more abundant, but with whales this may simply be a matter of how much biological material sloughed off of a single individual. It would be difficult to use eDNA as a proxy for whale abundance. Since it is ambiguous what "relative magnitude" means in this paragraph, perhaps this could be re-worded to emphasize co-occurrence.

-Are there any indications of the presence of any metazoan groups that are difficult to track by eye? Rare fish or squid? One can track seal or whale populations easily enough using traditional methods, but if eDNA was able to identify the presence of some cryptic groups that are hard to spot, it might be particularly interesting and useful for conservation methods.

-Since the paper is heavily focused on using WGCNA in time-series analysis and regression of the cluster profiles to environmental parameters, it would be appropriate to cite some of the first papers that did this in marine systems (doi:10.1038/ismej.2015.221, doi:10.1038/s41564-017-0008-3). Guidi et al is currently the only reference given in this regard, but to my knowledge it was actually the DeLong group that first started applying these statistical bioinformatic methods to marine systems (first report doi:10.1073/pnas.1502883112, though the references above would be more appropriate here). This would strengthen the paper since it would show that these bioinformatic methods have been used successfully in other systems (albeit with microbial groups only) and thereby reinforce the validity of the findings.

-The last paragraph of the Discussion is a bit vague and I think it could emphasize the implications of this work a bit more, perhaps with some citations that provide more detail. A revision here would be helpful.

Reviewer #2:

Remarks to the Author:

This study by Djurhuus ^[1]_{SEP} et al. uses eDNA sampled at a single location at multiple time points over 18 months to look at changes in marine communities, interactions among taxa within these communities and how these communities respond to environmental change. These are novel applications of eDNA, particularly the use of network analyses and comparison of eDNA signatures to environmental variables. This paper is important in that it moves eDNA studies from simply detecting taxa in a community to using eDNA to address key ecological questions such as assessing community turnover across environmental gradients, a key component of community ecology and ecological monitoring. These data provide meaningful community ecology data to marine ecosystems with the potential to enhance marine ecosystem monitoring efforts.

The manuscript is well written and easy to follow. It is clear from reading this paper that the authors have undertaken a tremendous amount of field, laboratory and analytical effort and have taken careful consideration of the multiple biases associated at each of these steps. However, there are a couple of areas of concern that should be addressed, as well as a number of smaller issues, outlined below.

One major concern is the assumption that taxonomic biases in PCR are consistent, as noted in lines 89-91 "This method assumes that amplification bias arises from template primer interaction, and that for any ^[1]_{SEP} given taxon-primer pair, this interaction is constant across samples allowing us to infer relative ^[1]_{SEP} changes in abundance between different taxa." While I understand the rationale behind scaling the abundance of sequence reads, I am concerned about the underlying assumption. Has this assumption been tested? I can see how PCR bias could change dramatically based on the taxa present and the concentration of that template DNA.

Another larger issue is that the analyses only classified OTUs to family or higher order levels of taxonomy. Many commonly available sequence classifiers provide lower level taxonomic resolution, including down to species level, including SINTAX, BLCA, RDP, Qiime SKlearn, Metaxa2, etc. and can be adapted to work on multiple metabarcoding loci. Even BLASTn + MEGAN6 LCA classification can provide species level taxonomic assignments at confidence of >80% identity. The conservative choice to classify OTUs to the level of taxonomy limits the interpretation of eDNA community assemblage results given the important key functional groups likely lost in the collapsing of OTUs to higher taxonomic levels. For example, multiple species of Balaenopteridae inhabit the Monterey Bay which each have distinct trophic ecologies, feeding strategies, migration patterns, etc. Collapsing classification to such a high taxonomic level obfuscates important ecological processes likely detected through eDNA metabarcoding. Furthermore, the authors stress the importance of eDNA for monitoring of marine ecosystems and conservation efforts. However, conservation efforts in the U.S. and globally are largely focused at the species or population level, thus by focusing on higher order taxonomy, the community assemblage data provides cursory related data to conservation efforts at best. The results of this study would be greatly enhanced using more accurate taxonomic assignment measures, either by including lower level taxonomic assignments or adopting a more powerful and better performing classifier, thereby allowing for the authors to engage in a much more nuanced and detailed analysis of community assemblage dynamics in the Monterey Bay.

Other issues:

Given the recent attention to the importance of the "max_target_seqs" parameters of BLAST highlighted by Shah et al. 2018 (Misunderstood parameter of NCBI BLAST impacts the correctness of bioinformatics workflows, *Bioinformatics*, bty833, <https://doi.org/10.1093/bioinformatics/bty833>), it is important to include the value for this parameter in Extended Data Table 1. It appears the standard value for the banzai pipeline is n=500 from the banzai GitHub page, but it is unclear what parameter

setting was used in this manuscript.

In the methods there was no mention of using negative controls: field blanks, extraction blanks, PCR-blanks, etc. Were there negative controls used in this study? The methods were unclear if such controls were used and this information is important for identifying sources of contamination and has become standard practice across eDNA metabarcoding studies.

Also, there can be significant variation among individuals eDNA samples, yet it seems like only a single sample was taken at each sampling even. Were any replicate sampling done to look at inter-sample variation? If not, how might inter sample variation influence the results and interpretations?

Line 56. eDNA is fairly widely used at this point. I'm not sure it's appropriate to describe this technique as not common practice.

Line 62-73 This section recapitulates the abstract, which isn't necessary

Figure 2. A and B are not particularly effective and really just seem like a big jumbled mess. C and F are more convincing and easily interpretable.

211-219. The implication here seems to be that these highly connected taxa are keystone species, although some of these taxa would seem to be really surprising as keystone species. There are two ways to interpret this result. First, that we've overlooked the importance of these keystone taxa, or two, perhaps we are being misled by these analyses. It would be worth commenting on this more.

225-230 Were these analyses done for the total data set? Or were they conducted independently for each season in the different years? It would add power to the network analyses to see the same patterns are observed independently in each of the years. While doing these analyses separately for each year isn't possible for all seasons for both years given the 18 month sampling, it is certainly possible for some seasons for both years.

283. This is a really challenging sentence that needs to be reworked

Reviewer #3:

Remarks to the Author:

When I agreed to review the Djurhuus et al. manuscript, I wasn't sure what to expect, since I predict that there will be a swathe of papers like this emerging over the next few years. The ability to multiplex different meta-barcoding markers across the same environmental samples should yield real power into deriving insights into ecological interactions and this is at the heart of the paper, produced by a renowned and respected group of investigators. Obviously, integrating a timeseries into the ecological analysis is really attractive, although it is disappointing that the team only managed to analyse eight non-replicated time points, but we all have to make difficult decisions in delivering the research programmes that we are involved in. Nevertheless, the team do deliver in generating a lot of data for a large number of taxa spanning multiple trophic levels and I really enjoyed seeing how they analysed and presented the data, giving real inspiration for future projects. The manuscript was very well written and summarises co-occurrence networks, akin to microbial/viral style microbiota analyses and then the team discuss possible ecological interactions. With measures of association as they have generated, the networks will comprise taxa that will occupy the same sort of ecological niche, but you also expect a number of them to interact directly. The team do not have any existing molecular gut content analysis to back-up their findings, but present interesting stories of strongly associated taxa;

some of which I'm not entirely sure make sense? Further, when looking at some of their correlated analyses in the latter figures, I am concerned that some of the stronger measures of association could be dictated by outlier events. When performing PLS analyses you can obviously derive significance above a certain threshold, but what happens when you remove a strong outlier? I will summarise further comments below and hope that the team will be able to address them in a future iteration of the manuscript but will for the time being, congratulate them on a very respectable submission of an interesting dataset. I like their work, but I would encourage the team to consider seriously at this stage as to whether they can make concrete assertions based on the limitations of the sampling design and potential issues with the data?

Major

Paragraph 231-239 - Can anything further be discussed regarding the co-occurrences. I appreciate that there are unlikely to be direct links between the polychaetes and the humpback whales, but it would be good to have a little more ecological background information?

Figure 3c - If you remove the seven potential outliers that are associated with higher levels of chlorophyll a, is this negative association significance, or relevant? Higher levels of chlorophyll a could be associated with PCR bias, due to inhibition via excessive amounts of organic material? Can the team reassure the readers that the samples have somehow not been affected by bias?

Is a sample size of 8 enough for valid statistical analyses? Many of the associations presented in Figure 8 are driven by outliers – what happens if you remove the outlier? N=30-50 would be a safer bet for these sorts of analyses.

Line 548-550 – Do the analyses look the same if deriving metrics based on rarefied recounts or simple proportions? I'm not suggesting re-analysing the whole dataset, would be curious to see if there were strong measures of association between the metrics employed and those that are commonly used in amplicon studies?

Do the team have any historical data to back up their findings?

Correlation and Correction for Multiple Comparisons – have these methods been employed/tested before, or have they been created for these data? Can they be trusted, or tested on existing mock datasets?

Medium

348 unique taxa seems quite low for bacteria, euks etc.? Can they comment?

Have they seen: Marine environmental DNA biomonitoring reveals seasonal patterns in biodiversity and identifies ecosystem responses to anomalous climatic events from Mike Bunce's group in PLoS Genetics - would be worth citing.

Line 121-123 - Can the team use any historical data to elucidate further?

Fig. 1a - I am a bit confused - how can the clustering of Tau correlations yield the hierarchical clustering, but the different subnetworks are paraphyletic in the visualization?

Fig. 2 - Lots of writing on figure (pie charts) too small.

Line 215-219 - The environmental DNA signal from different groups of organisms can come from living taxa, but also dead and decaying populations of organisms that have annual life history cycles. Can anything further be added here to suggest what ecological dynamics could be driving the patterns that are observed, according to knowledge of life history characteristics?

The team discuss in a very positive fashion all the taxa that they have discovered and monitored, what about the taxa that the meta-barcoding will miss? I think that a sentence or two making this caveat would make a more balanced perspective, unless they know they have 100% coverage of all the taxa in the Bay?

Line 318-319 - This sentence looks like it has been written as a flourishing remark, without too much consideration about the contents of the paper and would suggest a refresh.

Labwork - I cannot see information about the precise details as to how individual samples were labelled according to the molecular biological workflows that the team employed and this needs to feature in the manuscript please.

MEGAN is not infallible and given the small amount of OTUs generated, can the team confirm that they double checked the annotations manually?

Minor

Line 54 - please add broader reference (e.g. Deiner et al. MEC 2017)

Line 69-72 - Have another look at this sentence. Suggestion: Through network analysis, we found that groups of co-occurring organisms spanning different trophic levels were directly correlated to changes in environmental parameters, providing novel insights into the underlying response of whole communities to the environment and highlighting potential trophic interactions.

Figure 3 - I like the way to visualise this!

Line 283-284 - Is "but" needed in this sentence?

Figure 6 SST is too faint.

Figure 8 with Fig. 7 references?

Response to reviewers

Reviewer #1:

1. -One issue I am wondering about. The abundance of sequences in a given sample will be related to abundance for microbial taxa, but not for metazoans. One line 301 the authors state that baleen whales and Rhodospirillaceae are highly connected, and therefore likely present “in a similar relative magnitude”. In my mind this must mean that the two organisms co-occur in that sample, but nothing more. If one finds a large number of Rhodospirillaceae sequences in a library one can usually assume (not always accurately) that they are more abundant, but with whales this may simply be a matter of how much biological material sloughed off of a single individual. It would be difficult to use eDNA as a proxy for whale abundance. Since it is ambiguous what “relative magnitude” means in this paragraph, perhaps this could be re-worded to emphasize co-occurrence.

This has been reworded and “relative magnitude” has been deleted from the manuscript and exchanged with “patterns of abundance”. The reviewer makes a good point that for single celled organisms abundance can be inferred from sequencing. However, it has also been shown that amount of sequences is proportional to biomass, see refs Djurhuus et al. 2018, Kelly 2017, and Andruszkiewicz 2017. Which is why we chose to refer to amount of sequences as patterns of abundance.
2. -Are there any indications of the presence of any metazoan groups that are difficult to track by eye? Rare fish or squid? One can track seal or whale populations easily enough using traditional methods, but if eDNA was able to identify the presence of some cryptic groups that are hard to spot, it might be particularly interesting and useful for conservation methods.

We added this sentence to the discussion: “Using eDNA can also help identify cryptic taxa and detect taxa that are difficult to identify (e.g. the family Paralichthyidae (benthic flat fish), detected in this study), although it does not allow us to identify stages of life history and could include sequences from dead or decaying biological matter”. Albeit not necessarily hard to identify, we chose this benthic flatfish as an example due to the fact that these are hard to identify during dive surveys and require destructive sampling efforts to collect.
3. -cite (doi:10.1038/ismej.2015.221, doi:10.1038/s41564-017-0008-3). DeLong group that first started applying these statistical bioinformatic methods to marine systems (first report doi:10.1073/pnas.1502883112).

This is a good suggestion; these papers have been cited.
4. -The last paragraph of the Discussion is a bit vague and I think it could emphasize the implications of this work a bit more, perhaps with some citations that provide more detail. A revision here would be helpful.

We have tried to make the last paragraph of the discussion less vague according to these comments. “We conclude that when applied to an ecosystem over time, surveys using eDNA analysis can yield data-based biological indicators of

ecosystem change. Increasing such datasets can improve forecasting of biodiversity shifts that may result from environmental changes that occur across varied time scales and locations¹⁹. The essential biological observations provided by this technique will aid future efforts for proactive conservation of the life in the world's oceans.”

Reviewer #2 (Remarks to the Author):

5. One major concern is the assumption that taxonomic biases in PCR are consistent, as noted in lines 89-91 “This method assumes that amplification bias arises from template primer interaction, and that for any □ given taxon-primer pair, this interaction is constant across samples allowing us to infer relative □ changes in abundance between different taxa.” While I understand the rationale behind scaling the abundance of sequence reads, I am concern about the underlying assumption. Has this assumption been tested? I can see how PCR bias could change dramatically based on the taxa present and the concentration of that template DNA.

Since submission of this manuscript to Nature Communications, Dr. Ryan Kelly, has published a manuscript on the subject (testing the effects of PCR cycle numbers and primer amplification efficiency on the interpretation of relative/proportional abundance of eDNA, which now has been cited in the results and method sections. In this manuscript, we followed the recommendations for best practises of Kelly et al. 2019 for amplicon based sequencing (i.e. PCR protocol was consistent between samples and genetic markers we wanted to compare. The PCR cycle number was below 35. We did not compare absolute values of richness or Shannon indices, etc.)

6. Another larger issue is that the analyses only classified OTUs to family or higher order levels of taxonomy. Many commonly available sequence classifiers provide lower level taxonomic resolution, including down to species level, including SINTAX, BLCA, RDP, Qiime SKlearn, Metaxa2, etc. and can be adapted to work on multiple metabarcoding loci. Even BLASTn + MEGAN6 LCA classification can provide species level taxonomic assignments at confidence of >80% identity. The conservative choice to classify OTUs to the level of taxonomy limits the interpretation of eDNA community assemblage results given the important key functional groups likely lost in the collapsing of OTUs to higher taxonomic levels. For example, multiple species of Balaenopteridae inhabit the Monterey Bay which each have distinct trophic ecologies, feeding strategies, migration patterns, etc. Collapsing classification to such a high taxonomic level obfuscates important ecological processes likely detected through eDNA metabarcoding. Furthermore, the authors stress the importance of eDNA for monitoring of marine ecosystems and conservation efforts. However, conservation efforts in the U.S. and globally are largely focused at the species or population level, thus by focusing on higher order taxonomy, the community assemblage data provides cursory related data to conservation efforts at best. The results of this study would be greatly enhanced using more accurate taxonomic assignment measures, either by including lower level taxonomic assignments or adopting a more powerful and better performing classifier, thereby allowing for the authors to engage in a much more nuanced and detailed analysis of community assemblage dynamics in the Monterey Bay.

The BLASTn + MEGAN6 LCA did indeed yield species level assignment that we are confident in and have manually double-checked. However, for this study, we decided to analyse the biological community at a family level due to the sheer number of OTUs and genera observed. From the data after decontamination, occupancy modeling and quality filtering this study yielded:

12S: 167 OTUs, 30 genera, 32 species
16S: 3945 OTUs, 93 genera, 109 species
18S: 6600 OTUs, 701 genera, 997 species
COI: 7899 OTUs, 379 genera, 491 species

We have added a table (extended data table 2) listing the number of orders, families, genera, and OTUs for each genetic locus detected in this study. This table is based on the data prior to Kendall's Tau filtering since this step was only done after the data was already agglomerated at the family level. Hopefully this will give readers a sufficient overview of the data that we consolidated and analysed.

The reviewer makes an excellent point that each family can represent multiple species with differing ecological strategies. We could be missing taxa with differing ecologies, but within the megafauna it we only identified one species for each family (i.e. for Balaenopteridae, we actually only identified sequences down to humpback whale, no other Balaenopteridae species, however, we did identify sequences that were only annotated down to Balaenopteridae and not to species), showing an example of what could be done. To clarify, for some of the co-occurring taxa (Balaenopteridae, Otariidae, and Carangidae) we have added a parentheses containing species information (when the sequences grouped within that family only identified one species). Most commonly there was only one species identified within each family/taxa in the examples presented within this manuscript. We do suggest hypothesis testing and experimentation for conservation efforts, which would most likely be on a species level (for a much narrower group of target species, as opposed to a high-level ecosystem analysis presented here) but it is outside of the scope of this manuscript (see discussion second paragraph): “This property allows us to explore target species for conservation that could be secondarily combined with traditional methods (e.g. visual identification for abundance measures) for a more thorough survey of specific organisms or communities of interest. These analyses can predict taxon interdependencies that should be further investigated through hypothesis-based research and experiments.”

7. Other issues:

Given the recent attention to the importance of the “max_target_seqs” parameters of BLAST highlighted by Shah et al. 2018 (Misunderstood parameter of NCBI BLAST impacts the correctness of bioinformatics workflows, *Bioinformatics*, *bty833*, ¹), it is important to include the value for this parameter in Extended Data Table 1. It appears the standard value for the banzai pipeline is n=500 from the

banzai GitHub page, but it is unclear what parameter setting was used in this manuscript.

The value for max_target_seqs has been added to Extended Data Table 1.

8. In the methods there was no mention of using negative controls: field blanks, extraction blanks, PCR-blanks, etc. Were there negative controls used in this study? The methods were unclear if such controls were used and this information is important for identifying sources of contamination and has become standard practice across eDNA metabarcoding studies.

We did indeed include negative and positive controls in this study. Barplots of these have been added to the Extended Data, Fig. 3. See also methods section: decontamination of sequence data.

9. Also, there can be significant variation among individual eDNA samples, yet it seems like only a single sample was taken at each sampling even [sic]. Were any replicate sampling done to look at inter-sample variation? If not, how might inter sample variation influence the results and interpretations?

The reviewer is correct in that we only collected a single sample per sampling event. The team has previously observed that replication of samples for eDNA analysis within Monterey Bay, Florida Keys, and Puget Sound (see, Djurhuus et al. 2017, 2018, Kelly et al. 2017, 2018, and Andruszkiewicz et al. (2017)) yield very similar community assemblages between replicates. Although we agree that having replication of samples is best practice, we also believe that this would not have changed the outcome of this study and given the scale of the analysis chose to maximize the number of distinct samples processed. To rectify the lack of environmental sample replication, we did have PCR triplicates, which were all sequenced in order to observe the variation in taxa identified from a single sample.

10. Line 56. eDNA is fairly widely used at this point. I'm not sure it's appropriate to describe this technique as not common practice.

We have deleted this sentence.

11. Line 62-73 This section recapitulates the abstract, which isn't necessary

We recognize that some of the information in this section overlaps with the abstract, however we feel this paragraph adds important additional information to help the reader connect our methods and rationale for conducting this study.

12. Figure 2. A and B are not particularly effective and really just seem like a big jumbled mess. C and F are more convincing and easily interpretable.

Figure 2 A and B clearly illustrate the complexity of these networks. We used the networks and the centrality of taxa as a guide for which taxa to illustrate for figures C and F. Because of this we think it would be misleading to only show C and F without the full context of where those taxa fit within the networks.

Therefore, we kept A and B in the paper. We added the following to the caption: "These panels illustrate the complexity of the co-occurring taxa in the blue and

grey subnetworks, upon which the taxa representations of panels c) and f) are chosen.”

13. 211-219. The implication here seems to be that these highly connected taxa are keystone species, although some of these taxa would seem to be really surprising as keystone species. There are two ways to interpret this result. First, that we’ve overlooked the importance of these keystone taxa, or two, perhaps we are being misled by these analyses. It would be worth commenting on this more.

We revised the second paragraph of the discussion to address this concern. It now reads: “Disentangling the total biodiversity of a location can help identify previously unknown taxa that serve as keystone species, which in turn may be used as indicators of change due to external environmental or biotic factors...” & “These analyses can predict taxon interdependencies that should be further investigated through hypothesis-based research and experiments.”

14. 225-230 Were these analyses done for the total data set? Or were they conducted independently for each season in the different years? It would add power to the network analyses to see the same patterns are observed independently in each of the years. While doing these analyses separately for each year isn’t possible for all seasons for both years given the 18 month sampling, it is certainly possible for some seasons for both years.

All analysis was done for the entire dataset (clarified in the methods: weighted correlation network analysis) and not separated by season. We found that the years presented in this study are quite challenging to compare due to the fact that there is an El Niño occurring during one year of our sampling period (as pointed out in the methods). Thus, they would not be expected to be similar due to the distinct physical and chemical conditions observed during El Niño events and non-El Niño years in this region. We agree that additional sampling to explore inter-annual variation would be useful and hope that future efforts will address this. We added the following to the discussion to acknowledge this:

“Applying eDNA metabarcoding to monitor biodiversity in longer time-series will increase the power and validity of this technique for predicting ecosystem response to climate change and help inform conservation strategies.”

15. 283. This is a really challenging sentence that needs to be reworked

A revision has been made. “Pacific jack mackerel was a particularly important component of the California sea lion diet in 2015^{2,3}, whether their co-occurrence in our study is driven by direct predator-prey relationships or is instead a common response to an external driver is unclear.”

Reviewer #3 (Remarks to the Author):

16. The team do not have any existing molecular gut content analysis to back-up their findings, but present interesting stories of strongly associated taxa; some of which I'm not entirely sure make sense?

That is correct. We are merely trying to illustrate the complexity of the data and suggest taxa that might be interacting and some that are simply just present at the same time, likely affected in a similar way by their surroundings, see comment 18.

17. Further, when looking at some of their correlated analyses in the latter figures, I am concerned that some of the stronger measures of association could be dictated by outlier events. When performing PLS analyses you can obviously derive significance above a certain threshold, but what happens when you remove a strong outlier? I will summarise further comments below and hope that the team will be able to address them in a future iteration of the manuscript but will for the time being, congratulate them on a very respectable submission of an interesting dataset. I like their work, but I would encourage the team to consider seriously at this stage as to whether they can make concrete assertions based on the limitations of the sampling design and potential issues with the data?

We were also particularly concerned with this issue, and took care to address the question of false-positive correlations rigorously in our work. Most importantly for the reviewer's concern, we used Kendall's rank correlation as our pairwise statistic. Because this measure considers ranks rather than absolute values, it is highly robust to outliers (in a sense, there are no outlier values: the difference in rank between, say, the two highest data points is equal to that between the two lowest). In addition to choosing a statistic that effectively eliminated the outlier concern, we also used a highly conservative technique for correcting for false-discovery rates (Bonferroni). We are therefore able to be confident in our assessments of correlation, even given the small sample sizes.

Major

18. Paragraph 231-239 - Can anything further be discussed regarding the co-occurrences? I appreciate that there are unlikely to be direct links between the polychaetes and the humpback whales, but it would be good to have a little more ecological background information?

The exciting thing about this type of analysis is that it helps us uncover co-occurrence patterns that don't necessarily make sense based on ecological knowledge of how individual species interact. Unfortunately this is also what makes it challenging, because it's possible that there are not direct connections between the taxa, or that there are intermediary taxa driving the co-occurrence. Therefore it's important to use these data for further hypothesis-based testing to see if the co-occurrences are maintained over longer time scales and/or different locations, or to see if they correlate with any environmental parameters (i.e. are affected in the same way by the environment).

19. Figure 3c - If you remove the seven potential outliers that are associated with higher levels of chlorophyll a, is this negative association significance, or relevant? Higher levels of chlorophyll a could be associated with PCR bias, due to inhibition via excessive amounts of organic material? Can the team reassure the readers that the samples have somehow not been affected by bias?

It is not possible to quantify the PCR bias of our samples caused by chlorophyll. However, we do believe this could influence the proportional abundance of taxa, although our findings support findings of previous studies regarding presence of taxa, i.e. proportions of phytoplankton taxa present during the spring blooms.

The chlorophyll is just one environmental variable chosen as an example to illustrate correlation to the environment, and we still believe this correlation to be true. We have written this in the discussion:

“Amplification of DNA is subject to substantial primer biases and is almost certainly affected by inhibitors and variable abundances of template DNA; these complications likely affect the number of taxa detected with these methods, and consequently, we note that no method – including this one -- identifies all taxa in an ecosystem. However, eDNA analysis can combining observations across orders of magnitudes of organismal body size (i.e. microbes to mammals), across many trophic levels, and across vastly different domains of life.”

20. Is a sample size of 8 enough for valid statistical analyses? Many of the associations presented in Figure 8 are driven by outliers – what happens if you remove the outlier? N=30-50 would be a safer bet for these sorts of analyses.

The use of rank correlation avoids the outlier problem, and the conservative false-discovery correction gives us high confidence in the correlations despite the relatively small number of time points.

21. Line 548-550 – Do the analyses look the same if deriving metrics based on rarefied recounts or simple proportions? I’m not suggesting re-analysing the whole dataset, would be curious to see if there were strong measures of association between the metrics employed and those that are commonly used in amplicon studies?

We did, in fact, do this, but the results do not turn out very much different. The same 6 subnetworks were identified although some taxa were clustered in different subnetworks, the general trend was the same. In addition to this, we ran the networks focused only strongly positively correlated taxa, which yielded very similar results.

22. Do the team have any historical data to back up their findings?

We do not have any historical data of our own to support these findings, the only way we can back up our findings is by previously published studies in the area, which show similar results. These studies are referenced throughout the paper. The use of eDNA as we have used it here is new, so there is limited historical data available.

23. Correlation and Correction for Multiple Comparisons – have these methods been employed/tested before, or have they been created for these data? Can they be trusted, or tested on existing mock datasets?

It has never been done with eDNA methods in a natural system before. Recently a manuscript was published by Kelly et al. 2019, see references. These methods used a similar approach as was used here tested on a computer simulated mock community. We have followed the best practises published in the study by Kelly et al. 2019.

Medium

24. 348 unique taxa seems quite low for bacteria, euks etc.? Can they comment?

The 348 unique taxa are at the family level not genus or OTU, see methods “After decontamination we grouped all OTUs by their Family annotation or, if an OTU could not be annotated to Family, to order or class. In this manuscript we refer to a taxon as an individual Family, Order, or Class, using the most specific taxonomic resolution available at or above family level. One taxon might represent several OTUs, species or genera, but due to the sheer number of OTUs, species, and genera, all analyses was done at higher classifications to simplify the results. However, the analysis could just as easily be done at a lower taxonomic level, if desire”. This seems to be a reasonable number of unique families based on multiple studies, see Djurhuus et al. 2018, 2017, Sunagawa 2015, Andruszkiewicz 2017, and Sawaya 2019.

We have added an additional table; see Extended Data table 2 and response to reviewer #2 second concern (review 6).

25. Have they seen: Marine environmental DNA biomonitoring reveals seasonal patterns in biodiversity and identifies ecosystem responses to anomalous climatic events from Mike Bunce's group in PLoS Genetics - would be worth citing.

We were not aware of this paper and were not able to cite it since it was published after our submission of this manuscript. This is a great suggestion and has now been added to our manuscript.

26. Line 121-123 - Can the team use any historical data to elucidate further?

We have referenced other studies that have shown species richness changes, but we only have eDNA data for two years so far. There is definitely a potential for showing this in a future study, when we have a few years to base the observations on. In the discussion we have written “Applying eDNA metabarcoding to monitor biodiversity in longer time-series will increase the power and validity of this technique”.

27. Fig. 1a - I am a bit confused - how can the clustering of Tau correlations yield the hierarchical clustering, but the different subnetworks are paraphyletic in the visualization?

The reason the subnetworks are “paraphyletic” is explained in this blogpost by one of the creators (Peter Langfelder) of the WGCNA package: <https://peterlangfelder.com/2018/12/30/why-wgcna-modules-dont-always-agree-with-the-dendrogram/>.

The subnetworks are based on the hierarchical clustering (dendrogram Fig. 1) and subsequently a Dynamic Tree Cut, which identifies clusters as branches in the clustering tree. The reason for the dendrogram looking paraphyletic is because the:

“...Dynamic Tree Cut simply reflects the limitations of the dendrogram as a means of visualizing similarities of a large number of objects rather than something specific to Dynamic Tree Cut.”

And

“...once two clusters merge on the dendrogram, any points that are merged above that may belong to either of the two clusters, and won't form contiguous blocks of colors like the branches do. This also means that such outlying points cannot be assigned to clusters just based on the dendrogram; one needs the dissimilarity matrix to assign each point to its nearest cluster. This is what Dynamic Tree Cut does.”

We have added the link to this blog post in the methods section:

“We used the Dynamic Tree Cut function to divide the clusters into subnetworks, see: <https://peterlangfelder.com/2018/12/30/why-wgna-modules-dont-always-agree-with-the-dendrogram/>.”

28. Fig. 2 - Lots of writing on figure (pie charts) too small.
The writing on this figure has been enlarged for clarity.
29. Line 215-219 - The environmental DNA signal from different groups of organisms can come from living taxa, but also dead and decaying populations of organisms that have annual life history cycles. Can anything further be added here to suggest what ecological dynamics could be driving the patterns that are observed, according to knowledge of life history characteristics?
This is a great point that we focus on in the discussion, we have elaborated on this to say: “Using eDNA could also help identify cryptic taxa and detect taxa that are difficult to identify (e.g. the family Paralichthyidae (benthic flat fish)), it does not allow us to identify stages of life history and could include sequences from dead or decaying biological matter.”
30. The team discuss in a very positive fashion all the taxa that they have discovered and monitored, what about the taxa that the meta-barcoding will miss? I think that a sentence or two making this caveat would make a more balanced perspective, unless they know they have 100% coverage of all the taxa in the Bay?
We have added a sentence mentioning the limitations (due to low abundances, inhibitors, primer bias) of these methods in the discussion: “Amplification of DNA is subject to substantial primer biases and is almost certainly affected by inhibitors and variable abundances of template DNA; these complications likely

affect the number of taxa detected with these methods, and consequently, we note that no method – including this one -- identifies all taxa in an ecosystem.”

31. Line 318-319 - This sentence looks like it has been written as a flourishing remark, without too much consideration about the contents of the paper and would suggest a refresh.

Agreed. This sentence has been deleted and the paragraph revised.

32. Labwork - I cannot see information about the precise details as to how individual samples were labelled according to the molecular biological workflows that the team employed and this needs to feature in the manuscript please.

We assume the reviewer is referring to barcoding of the samples for sequencing. All barcodes have been added into a supplementary data file. Our apologies if we misunderstood the question.

33. MEGAN is not infallible and given the small amount of OTUs generated, can the team confirm that they double checked the annotations manually?

The team did indeed manually check the annotations - each annotation was checked by two independent team members. The taxa presented in the manuscript are all grouped at the family level and thus not represented at the OTU level. See response to reviewer #2, question 6, for more details and Extended Data table 2.

Minor

34. Line 54 - please add broader reference (e.g. Deiner et al. MEC 2017)

This reference has been added.

35. Line 69-72 - Have another look at this sentence. Suggestion: Through network analysis, we found that groups of co-occurring organisms spanning different trophic levels were directly correlated to changes in environmental parameters, providing novel insights into the underlying response of whole communities to the environment and highlighting potential trophic interactions.

Reviewer's suggestion has been added.

36. Figure 3 - I like the way to visualise this!

We thank the reviewer for this comment.

37. Line 283-284 - Is "but" needed in this sentence?

This sentence was revised.

38. Figure 6 SST is too faint [sic].

We revised the figure to make SST a darker grey color.

39. Figure 8 with Fig. 7 references?

This has been corrected

Reviewers' Comments:

Reviewer #4:

Remarks to the Author:

The manuscript explores the use of multilocus markers and environmental DNA (eDNA) methods to gain ecological insights across multiple trophic levels in an aquatic environment. The authors provide an interesting introduction on the use of eDNA metabarcoding to assess whole ecosystems and the importance of time series data to unravel complex biotic interactions related to ecosystem change. In addition, the methods and analysis are rigorous and discussions on potential avenues for further research are interesting and useful for the eDNA community.

1. line 585 – I am curious why the lower taxonomic information has not been analysed. I agree with review #2 that it should be included, even if it is placed in the SI. I think the lower taxonomic levels would provide much richer ecological information that is lost by collapsing the assignments into Family level and higher. If it is feasible to analyse this data, I would encourage the authors to include this data in their analyses. In addition to the ecological information that reviewer #2 commented on, I believe the eDNA community would benefit from a fuller understanding of how community network analysis can be used to analyse data derived from eDNA methods which in most cases is assigned to the lowest possible taxonomic level.

2. 595 – this text is difficult to grasp, perhaps illustrating how you calculated the taxon-specific index using an equation would help? It is unclear if the authors standardised taxa identified by more than one locus, using the same largest observed proportion for each taxon. Does line 602 answer my question? If so, I would not break the paragraph here as the previous paragraph only delivers half the information to the reader.

3. 670 – Further to reviewer # 3's question 19, it is still unclear to me if the co-authors looked for associations between taxa and environmental variables in winter and autumn subnetworks separately, analysed these two subnetworks together, and if summer and spring were also analysed together with autumn and winter? This detail should be included in methods for clarity. Furthermore, if only taxa from autumn and winter were analysed, how would the correlations be affected if the taxa sampled in summer and spring were included? For example, in line 117, the text states that a negative relationship is found with chlorophyll a in some taxa in autumn and winter. If only winter and autumn were included in the analysis, there may be some important information missing as conclusions are only based on a half the dataset. Post hoc analysis of autumn and winter subnetworks are certainly informative, but I would also be interested to see how the correlations change when all the data are included and the entire range of environmental variables are analysed. Could Fig 3b and 3c be colour coded to include correlations across all subnetworks? Or can this be included in Supplementary figure 8?

4. 260 – I am uncertain about what "that taxon" is referring too? Can the authors revise for clarity?

5. 294 – use combine instead of combining.

6. 320 – edits don't quite make sense. Suggest changing "is most likely to be presenting similar" to: "is more likely to present similar".

7. 326 – Can the authors make it clearer what is suggested in the cited research? The sentence is too vague. "suggested for microorganisms by Banerjee et al. 2018"

8. 333 – Suggest changing "monitoring of indicator taxa" to "monitoring indicator taxa"

9. 583 – all analyses were done.

10. Update citation of Kelly et al 2019 – now published in scientific reports.

<https://www.nature.com/articles/s41598-019-48546-x>

1. line 585 – I am curious why the lower taxonomic information has not been analysed. I agree with review #2 that it should be included, even if it is placed in the SI. I think the lower taxonomic levels would provide much richer ecological information that is lost by collapsing the assignments into Family level and higher. If it is feasible to analyse this data, I would encourage the authors to include this data in their analyses. In addition to the ecological information that reviewer #2 commented on, I believe the eDNA community would benefit from a fuller understanding of how community network analysis can be used to analyse data derived from eDNA methods which in most cases is assigned to the lowest possible taxonomic level.

We have added a supplemental figure analysing all the species-level assignments within the grey and blue subnetworks (see Supplemental Information, Figure 11). These networks have been visualized based on analysis at the species level. The results on a species level are very similar to the results on a family level. However, we are more confident in the family level results and believe they are more tangible to analyse, as well as easier to comprehend, due to the smaller amount of taxa to handle. In addition, the species level assignment excludes some OTUs that are not assigned to species level, which are included at the family level, see l. 309-312 and 621-623. By including this detailed analysis (Supplementary Information Fig. 11), we demonstrate the possibility of a species-level analysis, especially for the potential of using these types of analyses for conservation purposes. However, we believe the family-level assignments are more appropriate for the cross trophic level analyses for the purpose of demonstration of these results for this manuscript.

2. 595 – this text is difficult to grasp, perhaps illustrating how you calculated the taxon-specific index using an equation would help? It is unclear if the authors standardised taxa identified by more than one locus, using the same largest observed proportion for each taxon. Does line 602 answer my question? If so, I would not break the paragraph here as the previous paragraph only delivers half the information to the reader.

We have added an explanation and the equation for these calculations as referenced from Kelly et al. 2019. See l. 636-640.

3. 670 – Further to reviewer # 3's question 19, it is still unclear to me if the co-authors looked for associations between taxa and environmental variables in winter and autumn subnetworks separately, analysed these two subnetworks together, and if summer and spring were also analysed together with autumn and winter? This detail should be included in methods for clarity. Furthermore, if only taxa from autumn and winter were

analysed, how would the correlations be affected if the taxa sampled in summer and spring were included? For example, in line 117, the text states that a negative relationship is found with chlorophyll a in some taxa in autumn and winter. If only winter and autumn were included in the analysis, there may be some important information missing as conclusions are only based on a half the dataset. Post hoc analysis of autumn and winter subnetworks are certainly informative, but I would also be interested to see how the correlations change when all the data are included and the entire range of environmental variables are analysed. Could Fig 3b and 3c be colour coded to include correlations across all subnetworks? Or can this be included in Supplementary figure 8?

All data were analysed simultaneously. This text has been clarified on lines 111-120. The natural separation of the winter and autumn subnetworks was based on the presence/absence of the families (richness) within these networks, see Fig. 1b. We have added a figure in the supplemental information with the correlations of all subnetworks with the environmental parameters (see supplemental information, Fig. 5), which summarizes the correlations of the whole dataset with the environmental variables. In addition, we have changed supplementary information Figure 9 to show how all subnetworks compare to each other and their correlation to environmental variables.

Fig. 3b and 3c are only showing the connectedness of specific taxa and correlation between environmental parameters in the blue and grey subnetworks because these had strong correlations to specific environmental variables.

4. 260 – I am uncertain about what “that taxon” is referring too? Can the authors revise for clarity?

This has been clarified.

5. 294 – use combine instead of combining.

Done

6. 320 – edits don't quite make sense. Suggest changing “is most likely to be presenting similar” to: “is more likely to present similar”.

This change has been made.

7. 326 – Can the authors make it clearer what is suggested in the cited research? The sentence is too vague. “suggested for microorganisms by Banerjee et al. 2018”

This has been clarified.

8. 333 – Suggest changing “monitoring of indicator taxa” to “monitoring indicator taxa”

Done

9. 583 – all analyses were done.

Done

10. Update citation of Kelly et al 2019 – now published in scientific reports. <https://www.nature.com/articles/s41598-019-48546-x>

Done

Reviewers' Comments:

Reviewer #4:

Remarks to the Author:

I have read the revised manuscript and I am happy that all of my comments from the first round of review have been adequately addressed. However, one additional suggestion:

I like the addition of Figure 5 in the supplementary information. The text states that the blue and grey subnetwork have significant correlations with environmental parameters, but what about the rest of the subnetworks? If there are no other significant correlations I would suggest stating this in the legend for clarity.

Reviewers' Comments:

Reviewer #4 (Remarks to the Author):

I have read the revised manuscript and I am happy that all of my comments from the first round of review have been adequately addressed. However, one additional suggestion:

I like the addition of Figure 5 in the supplementary information. The text states that the blue and grey subnetwork have significant correlations with environmental parameters, but what about the rest of the subnetworks? If there are no other significant correlations I would suggest stating this in the legend for clarity.

This has been clarified.